# Guideline Adherence of Asymptomatic Bacteriuria Could Be Improved among General Practitioners in The Netherlands: A Survey Study

**DOI:** 10.3390/antibiotics11010075

**Published:** 2022-01-09

**Authors:** Tessa M. Z. X. K. van Horrik, Bart J. Laan, Tamara N. Platteel, Suzanne E. Geerlings

**Affiliations:** 1Department of Internal Medicine, Division of Infectious Diseases, Amsterdam University Medical Centers, Room D3-226 Meibergdreef 9, 1105 AZ Amsterdam, The Netherlands; b.j.laan@amsterdamumc.nl (B.J.L.); s.e.geerlings@amsterdamumc.nl (S.E.G.); 2Julius Center for Health Sciences and Primary Care, University Medical Center Utrecht, Room STR 7.111 Heidelberglaan 100, 3584 CX Utrecht, The Netherlands; T.N.Platteel-3@umcutrecht.nl

**Keywords:** asymptomatic bacteriuria, urinary tract infection, diagnostic stewardship, antimicrobial stewardship, general practice

## Abstract

Asymptomatic bacteriuria (ASB) is a common finding in certain populations. This study assessed general practitioners’ (GPs’) knowledge about ASB and their current clinical practice regarding urine testing. Methods: An online survey was used for GPs in the Netherlands from October to December 2020. Results: In total, 99 surveys were included in the analyses. All GPs strongly agreed with the statements about their knowledge and self-confidence regarding urine diagnostics and treatment of ASB. The median knowledge score was 4 out of 6 (IQR 2 to 6). Most GPs (64 of 92; 70%) followed the guideline for the choice of urine diagnostics and reported appropriate indications for urine testing. However, 71/94 (75.5%) GPs would treat patients for ASB if they have diabetes mellitus. Further, 34 (37%) of 92 participants would inappropriately repeat a urine test after a patient was treated for a urinary tract infection (UTI). One-third of the GPs responded that ASB was insufficiently addressed within the guidelines for UTI. Conclusion: These results indicate that knowledge about ASB could be improved in primary care in the Netherlands, mainly in diabetic patients that have ASB, as well as for follow-up tests after treatment for UTI.

## 1. Introduction

Urinary tract infections (UTIs) are among the most common infections worldwide. In the Netherlands, UTIs were one of the most common infections for which antibiotics were prescribed in primary care in 2018 [1]. As a result of the high incidence rates for UTI in primary care, it is indicated that urine tests (dipstick, microscopic analysis, dipslide, and urine culture) are frequently performed [2]. The UTI guidelines for general practice (GP) in the Netherlands recommend solely performing urine tests when a UTI is clinically suspected [3]. If the result of the dipstick test is inconclusive in a patient with a suspected UTI (i.e., negative nitrite and positive for leukocytes), an additional dipslide or microscopic analysis is recommended. In addition to this, these guidelines state that urine cultures should only be performed in patients with an uncomplicated UTI that are at higher risk of a complicated course of disease, if symptoms do not resolve after antibiotic treatment, or in case of a UTI while using antibiotic prophylaxis, and in all patients with a complicated UTI. Moreover, dipsticks may be performed for other indications, including renal insufficiency and albuminuria, which could lead to unintended positive urine dipstick results (i.e., positive nitrite and/or leukocytes) [4,5].

Asymptomatic bacteriuria (ASB) is defined as a positive urine culture obtained from a patient who does not have symptoms of a UTI [6,7]. One positive urine culture (≥10^5^ colony-forming units/mL) is sufficient to diagnose ASB. In certain populations, such as the elderly, patients with urinary catheters, and patients with diabetes mellitus, ASB is common [3,6]. Guidelines for the management of ASB state that patients with ASB should not be treated with antibiotics, except for patients with risk factors for a complicated UTI [3,6,8]. However, the results of several international studies showed that the prevalence of overtreatment of ASB still ranges from 45 to 83% in hospitals and nursing homes [9,10,11,12]. A prospective study that was performed in 10 nursing homes in the Netherlands in 2015 showed that 115 (32%) of 356 antimicrobial prescriptions for possible UTIs were inappropriate, and the main reason was ASB [13].

In the Netherlands, antibiotic use is low compared to other countries, but a large variation exists between the individual prescription rates by GPs [14,15]. A recent study investigated the use of three primary care guidelines and the motives for GPs to deviate from antibiotic treatment guidelines in the Netherlands [16]. The results of this study showed that antibiotics were prescribed for 72% of 26,017 UTI-related episodes that had an unclear indication for antibiotics. These indications were unclear, because the GP guidelines for UTI recommended three treatment options for healthy nonpregnant women with uncomplicated UTI (i.e., antibiotic prescription, a wait-and-see policy, or a delayed antibiotic prescription). The knowledge and management of ASB in primary care are not known. Since GPs diagnose and treat the majority of suspected UTIs, it is hypothesized that overtreatment of ASB is present in primary care. Therefore, this study aimed to investigate GPs’ knowledge about ASB and their current practice policies with regard to indication, choice, and follow-up of urine diagnostics. It was hypothesized that overtreatment of ASB was a result of an unnecessary performance of urine dipstick tests. This could be the result of performing urine dipsticks for inappropriate indications (e.g., routine testing or testing without knowing a patient’s symptoms). Hence, this could be improved as part of antibiotic stewardship.

## 2. Results

A total number of 132 primary care physicians responded to the survey invitations, of which 99 (75%) surveys (95 GPs and 4 GP residents) were included in the analysis after excluding surveys that lacked results on the statements or questions. The baseline characteristics of the participants are summarized in Table 1.

### 2.1. Knowledge and Perceptions with Regard to Asymptomatic Bacteriuria

The level of agreement was high for the statements regarding knowledge and self-confidence about appropriate urine testing and treatment of bacteriuria (Table 2). All participants strongly agreed with the statement ‘I know when to perform a urine test (urine dipstick, dipslide, or microscopic examination)’. Almost 80% of the GPs agreed with the statement that ASB does not harm patients in general. Interestingly, only 65% agreed with the statement that ASB does not harm their patients. Despite the presence of a special ASB chapter within the GP guidelines for UTI, one-third of the participants responded that these guidelines do not address ASB sufficiently.

A total of 94 (95%) participants answered the questions about which patients with ASB should be treated with antibiotics (Table 3). The median knowledge score was 4 (IQR 2–6). Sixty-eight percent of the participants earned a score of 4 or higher out of 6 on the knowledge questions, of whom 12% earned the maximum score (Appendix A). Most incorrect answers, 46% and 75%, respectively, were given to examples concerning patients with diabetes mellitus and ASB.

### 2.2. Reported Indications for Performing Urine Tests

Almost half of the participants responded that they would prescribe antibiotics for a patient with positive nitrite and/or leukocytes in the urine, regardless of or in expectation of the urine culture or dipslide results. Further, one-third of the participants would perform urine tests for inappropriate indications (Table 4). For example, 32 (37%) of 87 GPs reported performing a dipstick to check a patient’s urine after antibiotic treatment for a UTI as an appropriate indication. However, only 10% agreed with the statement that a patient’s urine should be checked after antibiotic treatment for cystitis or UTI (Table 3). Moreover, three-quarters of the GPs reported that they would perform a urine test at the patient’s request. Most reported ‘other indications’ (*n* = 5) for urine tests were appropriate, including albuminuria and pain in the lower abdomen (Appendix A).

### 2.3. Current Practice Concerning the Choice of a Urine Test and Follow-Up of Urine Diagnostics

In the Netherlands, urine dipsticks are available in almost all general practices. Next to this, a urine dipslide is often used by GPs. A dipslide is a test with two different agars (culture media) on each side to grow uropathogens in 24–48 h. If GPs aim to obtain a urine culture from a patient, they can send a urine sample to a regional microbiology laboratory, which is often located in a hospital. The microscopic analysis is a test that can be used to detect leukocytes and bacteria in patients’ urine and is not often used by GPs in the Netherlands.

In almost 70% of the general practices, patients’ symptoms were documented when a urine test is requested and the first diagnostic step for a suspected UTI was a urine dipstick, which measures the presence of leukocytes or bacteria directly (Table 5). One-fifth reported that urine dipsticks in combination with a dipslide were used as the first diagnostic step. Further, the most common initial treatment policies were an antibiotic prescription through the GP’s assistant or practice nurse and a (phone) consultation in which multiple treatment options were presented to the patient. From the participants who reported that another policy is followed, most reported that the medical policy depended on the patients’ symptoms, medical history, and comorbidity (Appendix A).

### 2.4. Reported Indications for Antibiotic Prescriptions

The majority (79–97%) of the participating GPs reported that they would prescribe antibiotics for appropriate indications, namely, an abnormal urine test result in combination with the patient’s urogenital symptoms (Table 6). However, concerning inappropriate and uncertain indications for antibiotic prescriptions, 17 (17%) of the GPs agreed with the behavior statement ‘I usually prescribe antibiotics if patients with a positive result on the urine dipstick request this, regardless of the presence of urogenital symptoms of these patients’ (Table 2). Following the reported indications for antibiotic prescriptions, we found that there were 12 (14%) participants who would prescribe antibiotics on patients’ request (Table 6). Further, 11.5% of the GPs reported they would prescribe antibiotics when a patient’s urine had changed in color, aspect, or smell. Other reported indications for prescribing antibiotics are listed in Appendix A.

## 3. Discussion

In this study, GPs felt confident about their knowledge about urine diagnostics and treatment of ASB. However, only two-thirds of the participants earned a score of 4 or higher out of 6 on the knowledge score and the majority of the participants would inappropriately treat a patient with diabetes mellitus and ASB. Furthermore, most GPs stated that they followed the GP guidelines for UTI concerning the choice of the diagnostic test. Remarkably, over one-third of the GPs would inappropriately perform a urine test to check a patient’s urine after antibiotic treatment for UTI. Most GPs reported appropriate indications for antibiotic prescriptions. Nevertheless, at least 10% of the GPs would prescribe antibiotics at patient’s request, regardless of the presence of UTI-related symptoms.

### 3.1. Comparison with Existing Literature

The results of this study indicate that GPs overestimated their knowledge and behavior, which could be due to the social desirability bias [17]. These findings also correspond with previous research that found that people, including doctors, have a limited ability to accurately self-assess their performance [18,19]. Interestingly, the majority of the participants in this study agreed that ASB does not harm patients in general, yet fewer participants agreed to this statement when it addressed their patients (80% and 63%, respectively). This could be explained by physicians considering their patients more vulnerable than other patients, which was also found in a semi-structured qualitative interview study about attitude and perspectives for prescribing antibiotics, performed in four hospitals in the United Kingdom [20].

The knowledge questions were answered adequately (score 4 out of 6) by 68% of the participants. However, one-third of the participants scored moderately (score < 4 out of 6). A possible explanation for these moderate results is that ASB is insufficiently addressed in the GP guidelines for UTI, which is also indicated by one-third of our participating GPs in the results. We hypothesize that this perception could be because ASB is only mentioned and explained in the complete version of the guideline and not in the summary [3]. Further, in a survey study performed among medical specialists and residents in the USA in 2014, it was shown that the acceptance of ASB guidelines was high, but healthcare workers’ knowledge of its content and their ability to apply the guidelines to real patients were lacking [21]. This could clarify why one-third of the GPs earned moderate knowledge scores.

Almost three-quarters of the GPs would perform a urine test at a patient’s request. The results of a previously performed qualitative study in the Netherlands showed that patients’ clinical condition and comorbidity are the most important influencing factors in performing microbiological diagnostics tests for physicians and barriers to the implementation of clinical practice guidelines [22]. The results of multiple qualitative studies that investigated influencing factors of antibiotic prescribing have highlighted time as an important contributor [16,23]. It is likely that the GPs in this study may also have agreed with a patient’s request for antibiotics due to a lack of time to discuss with the patient about the symptoms and indications for antibiotic treatment. In addition, the results of previous studies have shown that GPs assumed that patients require antibiotics if they have an infection [24], yet patients also have a need for advice and reassurance [25,26]. This could also explain why at least 10% of the GPs would prescribe antibiotics for patients with ASB. Nevertheless, antibiotic treatment for ASB is in general unnecessary [3,6]. Moreover, overtreatment of ASB could lead to an increase in antimicrobial resistance and *Clostridium difficile* infections [6,27]. Concerning the hypothesis that GPs would inappropriately treat ASB because of unnecessary performance of urine dipsticks, the results of this study do not clarify if GPs performed dipsticks routinely as was found in studies that were performed in hospital settings [28,29].

### 3.2. Strengths and Limitations

One of the strengths of this study is that only a few studies have examined GP’s knowledge and management of ASB thus far [30]. Further, the Dutch GPs have very high professional standards with strong implementation of guidelines, and they are responsible for the majority of all patientcare in the Netherlands. Therefore, it is especially interesting to get insight into the current clinical practice regarding urine diagnostics and ASB in primary care. We will use these findings in the development of strategies to reduce inappropriate urine diagnostics and overtreatment of ASB in the Netherlands.

Nevertheless, this study had several possible limitations. First, the total number of invited physicians was unknown, and we were unable to calculate a response rate because of our recruitment strategy. The response rate is, however, estimated around 10% because at least 900 GPs were invited to participate in this survey. Second, demographics (e.g., working area, village, or city) were not included in the baseline characteristics to shorten the time to complete the survey but the age and years of work experience were included. We believed these factors would be more relevant concerning familiarity with guidelines and practice policies. Regardless, we consider the data representative for GPs in the Netherlands because a random sample was invited. Third, most GPs reported that patients’ symptoms are documented when a urine test is requested and thereby suggested that urine tests are not performed in asymptomatic patients. However, the results of this study do not clarify whether and which particular symptoms are taken into consideration before a urine test is performed. Fourth, common limitations for survey studies are selection bias and social desirability bias. In this study, the selection bias could be caused by the participation of GPs who were already interested in the subject and could have been more willing to participate. However, both GPs with more and less knowledge about ASB could have participated (e.g., to gain more knowledge) that could have led to either an overestimation or an underestimation of these results. We believe that the social desirability bias is limited because all data were anonymously collected.

### 3.3. Implications for Further Research

The results of this study suggest that guideline adherence for the management of ASB could be improved in primary care in the Netherlands. This could also be applicable to other countries because in general, the Dutch GPs are reluctant to use antibiotics compared to others [31]. Improving this guideline adherence is especially recommended for patients with diabetes mellitus and ASB since this is one of the top healthcare problems (103 per 1000 registered patients) in primary care in the Netherlands [2]. Moreover, the prevalence of ASB is high among patients with diabetes mellitus [32]. The results of this study do not clarify whether GPs would quit the initiated antibiotic treatment when a urine culture or dipslide result turned out negative. Additionally, patients’ perspectives on urine tests and antibiotic treatment for UTI or ASB were not included in this study, but this will be considered in future research. These perspectives could be relevant since at least 10% of the GPs reported they would prescribe antibiotics at patients’ request. Moreover, patients who have experience with UTIs could have multiple motivations for requesting urine tests, such as fear of a complicated UTI [33,34]. With regard to antibiotic stewardship, it could also be interesting to focus more on non-antibiotic alternatives to treat symptomatic UTIs [35,36]. At this moment, we are investigating the management of UTIs in combination with shared decision making in the GP office. However, ASB does not require treatment in most cases. Therefore, this topic was not included in this study. Next to this, we are currently investigating the overtreatment of ASB with a multicenter study in the emergency department in the Netherlands [37]. In order for inappropriate diagnostics and overtreatment of asymptomatic patients or patients with non-specific urogenital complaints to be reduced, ASB could be addressed within the summary of the guidelines for UTI. Likewise, more attention could be paid to ASB through online training programs for GPs. Educational programs should also focus on diagnostic stewardship apart from improving antibiotic prescriptions because positive urine test results could lead to inappropriate antibiotic prescriptions [11,38].

## 4. Materials and Methods

### 4.1. Study Design and Setting

An online survey study was performed among GPs and GP residents in the Netherlands from October to December 2020. A random sample of 900 general practices was drawn from a database of 5000 general practices by the Netherlands Institute for Health Services Research. The GPs working in these general practices were invited to participate in this online survey through post or e-mail. An e-mail reminder for participating in this survey was sent after six weeks. Additionally, GPs and GP residents were invited to participate in this survey through social media and personal networks of the participating GPs. All participants received an invitation in which the aim of the study and content of the survey were briefly described. A quick response (QR) code to the online survey was included in the invitation. Participants had to give informed consent and there was no incentive for completing the survey. Participation in the survey was anonymous, and participants were able to quit the survey anytime.

### 4.2. Survey and Data Collection

Since no validated survey to investigate the presence and management of ASB in primary care existed, an open electronic survey for physicians in the Dutch language was developed in collaboration with two authors (T.P. and S.G.) of the GP guidelines for UTI [3]. The survey consisted of 22 items on 8 pages and took 5 to 10 min to complete. LimeSurvey version 2.6.7 was used to distribute the surveys and collect the data. All questions were mandatory and could not be skipped. Adaptive questioning or randomization of the items were not applied. Participants were able to review and change their answers. Since this study used an open survey and participation was completely anonymous, no personal data, such as unique visitor rate or registration, were tracked.

The content was divided into three parts: (1) baseline characteristics of the participating GP; (2) statements and knowledge questions about ASB; and (3) the current local practice concerning indication, choice, and follow-up of urine diagnostics. The statements and knowledge questions were based on a previously reported survey that was developed to investigate knowledge of ASB, the management of catheter-associated-UTI, the association with work experience, and familiarity with the Infectious Diseases Society of America guidelines on ASB [21]. Additionally, statements that focused on a primary care setting were conducted in close collaboration with authors of the GP guideline for UTI (T.P., S.G.) [3]. For all statements, a 5-point Likert scale ranging from strongly disagree to strongly agree was used. For the knowledge questions, each correct answer was rewarded with one point and a score of 4 out of 6 points was considered adequate. The indications for performing urine tests and prescribing antibiotics were based on the results of a previous study that investigated risk factors and barriers associated with inappropriate diagnostics and treatment of urinary tract symptoms and ASB [11]. The survey was not formally validated but was proofread by colleagues. The Checklist for Reporting Results of Internet E-Surveys (CHERRIES) was used to report the study [39] (Appendix A).

### 4.3. Statistical Analysis

In total, 33 surveys that lacked relevant data were excluded (e.g., only information about baseline characteristics and if no informed consent was given). Incomplete questionnaires that were terminated early were included in the analyses if at least the statements were answered. The categorical data were presented as frequencies and percentages. Descriptive statistics were used to establish the median knowledge score. For the knowledge questions, all correct answers were awarded with one point with a maximum score of 6 (range 0–6). One exemplary case of the knowledge question was excluded from the analysis since both ‘yes’ and ‘no’ could be considered correct answers. Statistical analyses were performed using IBM SPSS Statistics for Windows, version 26.0 (IBM Corp., Armonk, NY, USA).

## 5. Conclusions

In summary, this study provided insight into the knowledge and self-reported current practice concerning urine diagnostics and asymptomatic bacteriuria in primary care in the Netherlands. The results of this study showed that most GPs follow the GP guidelines for UTI regarding the initial diagnostic test in case of a suspected UTI. Nevertheless, there is room for improvement in the practical application of these guidelines, especially with regard to the treatment of ASB in patients with diabetes mellitus.

## Figures and Tables

**Table 1 antibiotics-11-00075-t001:** Baseline characteristics of participating general practitioners.

**Age Group (*N* = 99)**	***n* (%)**
≤35 years	13 (13.1)
36–45 years	39 (39.4)
46–55 years	28 (28.3)
>55 years	19 (19.2)
**Workplace * (*N* = 99)**	***n* (%)**
Solo practice	29 (27.1)
Duo practice	29 (27.1)
Group practice	27 (25.2)
Healthcare center	22 (20.6)
**Type of GP (*N* = 99)**	***n* (%)**
Practice owner	72 (72.7)
Employed by another GP	10 (10.1)
Permanent locum	13 (13.1)
Resident	4 (4.0)
**Work experience in years (*N* = 99)**	***n* (%)**
<3	5 (5.1)
3–7	22 (22.2)
7–10	11 (11.1)
>10	61 (61.6)

*n* are the number and percentage of respondents. GP: general practitioner. * In total, six participants were working in more than one workplace.

**Table 2 antibiotics-11-00075-t002:** General practitioners’ self-reported knowledge and perceptions regarding asymptomatic bacteriuria.

Statement	*N*	Strongly Disagree *n* (%)	Disagree *n* (%)	Neutral *n* (%)	Agree *n* (%)	Strongly Agree *n* (%)
**Knowledge and self-confidence**						
I know when I should perform a urine test.	99	0 (0.0)	0 (0.0)	0 (0.0)	13 (13.1)	86 (86.9)
I feel confident about interpreting the urine test results.	99	0 (0.0)	2 (2.0)	3 (3.0)	42 (42.4)	52 (52.5)
I am familiar with the concept ‘asymptomatic bacteriuria’.	99	0 (0.0)	1 (1.0)	3 (3.0)	28 (28.3)	67 (67.7)
I know when I should and when I should not treat bacteriuria.	99	0 (0.0)	5 (5.1)	5 (5.1)	53 (53.5)	36 (36.4)
**Behavior**						
I usually prescribe antibiotics to treat bacteriuria in patients with a positive nitrite and/or leukocyte result in urine, regardless of the urine culture or dipslide results.	98	7 (7.1)	26 (26.5)	19 (19.4)	30 (30.6)	16 (16.3)
I would rather treat ASB in an older patient than in a younger patient.	99	15 (15.2)	43 (43.4)	9 (9.1)	28 (28.3)	4 (4.0)
I would rather treat an older patient with non-specific urogenital complaints and bacteriuria with antibiotics than a younger patient with the same findings.	99	5 (5.1)	23 (23.2)	20 (20.2)	40 (40.4)	11 (11.1)
I usually prescribe antibiotics if patients with a positive result on the urine dipstick request this, regardless of the presence of urogenital symptoms of these patients.	99	12 (12.1)	51 (51.5)	19 (19.2)	14 (14.1)	3 (3.0)
The urine of a patient should be checked after antibiotic treatment for UTI.	99	22 (22.2)	53 (53.5)	14 (14.1)	6 (6.1)	4 (4.0)
**Social norms**						
In general, my colleagues treat a positive urine dipstick result with antibiotics.	99	0 (0.0)	11 (11.1)	20 (20.2)	51 (51.5)	17 (17.2)
In general, my colleagues believe that following clinical practice guidelines improves patient care.	99	0 (0.0)	1 (1.0)	25 (25.3)	48 (48.5)	25 (25.3)
**Risk perception**						
In general, ASB does not harm patients.	97	0 (0.0)	6 (6.2)	14 (14.4)	59 (60.8)	18 (18.6)
In general, ASB does not harm my patients.	97	0 (0.0)	16 (16.5)	18 (18.6)	53 (54.6)	10 (10.3)
**Clinical practice guidelines**						
The national GP guidelines for UTI addresses ASB sufficiently.	97	5 (5.2)	31 (32.0)	29 (29.9)	22 (22.7)	10 (10.3)
The national GP guidelines for UTI are easy to follow with regard to ASB.	96	3 (3.1)	20 (20.8)	40 (41.7)	19 (19.8)	14 (14.6)
There are other guidelines that interfere with the GP guidelines for UTI concerning ASB.	96	1 (1.0)	18 (18.8)	69 (71.9)	7 (7.3)	1 (1.0)

*n* are number and percentage of the respondents. Urine test: urine dipstick, microscopic examination, dipslide, and urine culture. Positive urine test result: positive nitrite and/or leukocytes. ASB: asymptomatic bacteriuria; GP: general practitioner; UTI: urinary tract infection.

**Table 3 antibiotics-11-00075-t003:** Responses to asymptomatic bacteriuria cases.

Should This Patient Be Treated with Antibiotics for ASB *? (*N* = 94)	*n* (%) ** Respondents	*n* (%) Correct Answers
**Patients with ASB who should not be treated with antibiotics**		
96-year-old female who lives in a nursing home	3 (3.2)	91 (96.8)
32-year-old male with insulin dependent diabetes mellitus	71 (75.5)	23 (24.5)
50-year-old female with diabetes mellitus without insulin dependency	44 (46.8)	50 (53.2)
57-year-old female with a urinary catheter and a positive dipslide result	9 (9.6)	85 (90.4)
**Patients with ASB who could be treated with antibiotics**		
28-year-old pregnant female	62 (66.0)	62 (66.0)
54-year-old female with recurrent UTI	21 (22.3)	21 (22.3)

* patients with an abnormal urine dipstick result, but without fever or urogenital symptoms. ** *n* are the number and percentage of respondents who would treat this patient with antibiotics. ASB: asymptomatic bacteriuria; GP: general practitioner; UTI: urinary tract infection.

**Table 4 antibiotics-11-00075-t004:** Reported indications for urine tests.

Indications for Performing Urine Tests * *N* = 87	*n* (%) **	*n* (%) Correct Answers
**Appropriate indications**		
Dysuria, frequent urination, hematuria, urinary urgency	86 (98.9)	86 (98.9)
Delirium	84 (96.6)	84 (96.6)
Urinary incontinence	68 (78.2)	68 (78.2)
Macroscopic hematuria	67 (77.0)	67 (77.0)
Patient with urogenital symptoms after recent use of urinary catheter	78 (89.7)	78 (89.7)
Patient with urogenital symptoms after recent UTI	84 (96.6)	84 (96.6)
Fever of unknown origin	80 (92)	80 (92)
**Inappropriate indications**		
Nausea/vomiting	4 (4.6)	83 (95.4)
Check after antibiotic treatment for UTI	32 (36.8)	55 (63.2)
**Indications that could be both appropriate and inappropriate**		
Patient’s request	65 (74.7)	n/a
Fever with a focus different than UTI	1 (1.1)	n/a
Overall malaise, fatigue, dizziness, syncope	13 (14.9)	n/a

* Multiple answers possible. ** *n* are the number and percentage of respondents who chose the indication for urine tests. n/a: not applicable; UTI: urinary tract infection.

**Table 5 antibiotics-11-00075-t005:** Choice and follow-up of urine diagnostics in current practice.

Local Practice Characteristics (*N* = 92)	Number of Respondents (%)
**What urine test is performed as initial diagnostic step?**	
Urine dipstick	64 (69.6)
Urine dipstick + microscopic analysis	8 (8.7)
Urine dipstick + dipslide	20 (21.7)
Dipslide	0 (0.0)
Urine culture	0 (0.0)
**Are patients’ complaints or symptoms documented when their urine is tested (e.g., by using a questionnaire)?**	
Yes	90 (97.8)
No	2 (2.2)
**When the urine dipstick has a positive result, is a dipslide or microscopic examination performed?**	
Yes	10 (10.9)
No	82 (89.1)
**When the urine test result is abnormal, what local policy is followed?**	
Patient receives antibiotics through the GP assistant or practice nurse	48 (52.2)
Patient always receives an appointment for a phone consultation with the GP	3 (3.3)
Patient visits the consultation hour and will be physically examined	6 (6.5)
Patient receives multiple treatment options, including antibiotic treatment, wait-and-see policy, or a delayed prescription	46 (50.0)
Other	21 (22.8)

*n* are the number and percentage of respondents. Urine test: urine dipstick, microscopic examination, dipslide, and urine culture. Positive urine test result: positive nitrite and/or leukocytes. GP: General practitioner.

**Table 6 antibiotics-11-00075-t006:** Reported indications for antibiotic prescriptions.

Indications for Antibiotic Prescriptions * *N* = 87	*n* (%) **	*n* (%) Correct Answers
**Appropriate indications**		
Abnormal urine dipstick result, dipslide, and/or microscopic examination	71 (81.6)	71 (81.6)
Dysuria, frequent urination, hematuria, urinary urgency	69 (79.3)	69 (79.3)
**Inappropriate indications**		
Changes in color, aspect, or smell of urine	10 (11.5)	77 (88.5)
Nausea/vomiting	2 (2.3)	85 (97.7)
**Indications that could be both appropriate and inappropriate**		
Altered mental status or behavior, other than delirium	20 (23)	n/a
Macroscopic hematuria	14 (1.61)	n/a
Patient’s request	12 (13.8)	n/a
Other	11 (12.6)	n/a

* Multiple answers possible. ** *n* are the number of respondents who chose the indication for antibiotic prescriptions. n/a: not applicable.

## Data Availability

Data are available upon reasonable request.

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
