# Peer review of "Guideline Adherence of Asymptomatic Bacteriuria Could Be Improved among General Practitioners in The Netherlands: A Survey Study"

_antibiotics, 2022, doi:10.3390/antibiotics11010075_

Round 1

Reviewer 1 Report

The authors focused on the Guideline adherence of asymptomatic bacteriuria could be improved among general practitioners in the Netherlands: a survey study 

Shape requests

The topic should be of interest but he quality of presentation is low, the neglected aspect of the work greatly decreasing its quality:

  • Please avoid using the personal manner of addressing "We' and "our" and replace it with the impersonal one. The English will sound much more professional. Please revise the entire manuscript in this regard.
  • Too many interspaces between the paragraphs - please remove them and use the inter spatiality recommended in the draft for Antibiotics, with no empty lines;
  • The tables are not properly arranged. Check the Instructions for authors https://www.mdpi.com/journal/antibiotics/instructions  and set them accordingly;
  • Text of the sections - justify;
  • References must be set also according to MDPI journals requests. Complete all information requested for each ref. Please see the same Instructions for authors. As an advice, from my experience: for all your papers consider also the content and the aspect - both are important in the presentation, and also the reviewers/editors consider them.

Content requests

As the last, separate paragraph of Introduction please make the aim of the study relevant. What makes special this study? Which is its novelty character or its special aspects? Why have the author chosen this topic? What differentiate this paper from others in the same topic? Actual text is not relevant at all in this regard, there are tenths/hundreds of papers in the same topic.

Discussion  section is poor. L 220, please discuss the implications of prescribing antibiotics in situations where they are not needed such as in ASB, and the impact of this practice in increasing the resistance to antibiotics - I suggest checking https://www.mdpi.com/2079-6382/9/2/81. Please discuss also certain educational measures in order for the GPs to increase the adherence to prescribing guidelines, and the role of certain alternative therapies for patients with ASB. I suggest checking and refer to: https://www.ncbi.nlm.nih.gov/pmc/articles/PMC7731396/ 

To enrich the content of this part, I suggest adding a separate paragraph regarding to the most effective / new treatments recommended and available for this disorder. Literature is plenty of papers in this direction as  I found very quickly few very different suggested treatments by specialists: https://doi.org/10.1016/j.ejps.2019.105067   https://doi.org/10.3390/molecules25235593 https://doi.org/10.3892/etm.2020.8664 I am sure that you can find more.

In the Methods section please make a graphic/flow chart about the study methodology that simplifies the inclusion/exclusion methods of the participant (based on which criteria, the respondents were chosen?) and how the GPs were interviewed. Moreover, please detail/clarify: who made the surveys? who validated them? there were some collaboration with sociologists, specialists in such questionnaires? were these questionnaires pre-tested before their application to all respondents? based on which criteria the items were chosen/ how do you have chosen/decided the optimal items? etc.

Reviewer 2 Report

The article “Guideline adherence of asymptomatic bacteriuria could be improved among general practitioners in the Netherlands: a survey study” is on a very interesting topic, overall the manuscript is well written but I have following comments/suggestions,

  1. The authors have stated that “A random sample of 900 general practices was drawn from a database of 5000 general practices by the Netherlands Institute for Health Services Research”, why they have used a random sample size of 900? Why haven’t they calculated the sample size?
  2. The authors have conducted an online survey and there are some checklists for conducting online surveys, kindly refer to “Checklist for Reporting Results of Internet E-Surveys (CHERRIES)”.
  3. It is not clear from the method section that how the study questionnaire was validated? Or it was previously validated? Which language was used for the survey? How was the internal consistency checked?
  4. What is the difference in responses between the GP’s and GP residents? It will be more appropriate to show this difference and add some lines in the discussion section as it will give an idea to the readers about the differences in these groups.

Reviewer 3 Report

First of all, thank you for the opportunity to review this interesting manuscript.

The paper evaluates the general practitioners' approach to asymptomatic bacteriuria (ASB) and the general approach to antibiotics in patients with urinary tract infections.

The work is well designed, the methodology is adequate, the results and discussions are clearly stated. I have only three small comments and after their solution I recommend the manuscript for acceptance.

  1. I recommend specifically explaining the term microscopic analysis of urine, resp. as implemented in GP practice in the Netherlands.
  2. The definition of ASB is given in the text, however, I recommend clarifying this. How is this diagnosis specifically determined, based only on the cultivation of urine in a microbiological laboratory? Is one cultivation is enough, or are repeated cultivations necessary?
  3. I consider it appropriate to supplement the information and availability of microbiological urine culture, resp. in microbiological laboratories, for GP in the Netherlands.

After resolving my comments I recommend the manuscript for acceptance.

Round 2

Reviewer 1 Report

The authors responded to all my requests.

Reviewer 2 Report

The authors have addressed all of my comments/suggestions in their revised submission.